# Curing Cancer: Lessons from a Prototype

**DOI:** 10.3390/cancers13040660

**Published:** 2021-02-07

**Authors:** Shi-Ming Tu, Louis L. Pisters

**Affiliations:** 1Department of Genitourinary Medical Oncology, The University of Texas MD Anderson Cancer Center, Houston, TX 77030, USA; 2Department of Urology, The University of Texas MD Anderson Cancer Center, Houston, TX 77030, USA; lpisters@mdanderson.org

**Keywords:** testicular cancer, personalized care, precision medicine, targeted therapy, cancer stem cell, second malignancy, somatic transformation

## Abstract

**Simple Summary:**

Germ cell tumor of the testis (TGCT) teaches us that to cure cancer, we need to acquire and apply proper biological insight and clinical acumen. In 1946, about 90% of patients with metastatic TGCT died within the first year of diagnosis. Today, over 90% of the same patients are curable. This complete reversal in the cure rate of TGCT is not because we have designed better drugs (we have not), but because we have learned how to use the same drugs in the right patients under the right settings. Importantly, TGCT is a prototype stem cell tumor that may hold the key to unlocking the origin of cancers, thereby enhancing our understanding of cancer and improving the cure and care of patients with cancer.

**Abstract:**

Germ cell tumor of the testis (TGCT) is a remarkably curable solid tumor even when it is widely metastatic and patently heterogeneous. It provides invaluable clues about the origin and nature of metastasis and heterogeneity, cancer dormancy and late recurrence, drug sensitivity and resistance, tumor immunity, and spontaneous remission that would enable us to enhance the cure and improve the care of patients with other currently intractable solid tumors. After all, germ cells are primeval stem cells and TGCT are a perfect stem cell tumor for us to investigate a stem cell versus genetic origin of cancer. In many respects, TGCT is a prototype stem cell tumor that will enable us to elucidate the role of differentiation versus dedifferentiation in the evolution of a complex mixed tumor. It will help us decipher relevance of the genome versus the epi-genome in a progenitor cancer stem cell versus a progeny differentiated cancer cell. Importantly, clarification of a cellular context versus the genetic makeup in cancer has immense clinical implications. We postulate a unified theory of cancer derived from seminal TGCT research to improve personalized cancer care. Contrary to current norms and conventional wisdom, we propose that when it concerns a complex rather than simple cancer and a mixed rather than pure tumor (which is practically all solid tumors) multimodal therapy trumps targeted therapy and integrated medicine overrides precision medicine.

## 1. Introduction

Germ cell tumor of the testis (TGCT) is a unique cancer. It is prominently curable, yet still potentially lethal. Genetically, it is relatively simple, but clinically, it can be deceptively complex [1]. Importantly, TGCT may hold the key to unlocking the origin of cancers, thereby enhancing our understanding of cancer and improving the care of patients with cancer.

In 1946, about 90% of patients died from their metastatic TGCT within the first year of diagnosis [2]. Today, we have cured over 90% of the same patients. Although chemotherapy did instigate a breakthrough and increase the cure rate of metastatic TGCT to about 60% in the 1970s, we have actually not designed any better drugs since then [3]. What are the evidence and experience that bring us to where we are today? What are the ideas and methods that will take us to where we need to be tomorrow?

In this article, we introduce the idea that TGCT is an ideal tumor model for us to interrogate the theory of cancer having a stem cell origin and the concept of cancer being a stem cell disease [4,5,6,7]. We illustrate that many cancer hallmarks, including heterogeneity and dormancy, are in fact stem-ness earmarks. During carcinogenesis, there is often reemergence and resurgence of embryogenesis. There is a reason and a purpose for the dictum “oncology recapitulates ontogeny”.

We reminisce on classic experiments of TGCT that confirm a stem cell theory of cancer. We revisit seminal experiments of other cancers supporting the idea that cancer is a stem cell disease. We reiterate recent experiments that affirm a close relationship between a primary TGCT and a second subsequent malignant neoplasm (SMN) separated by years, if not decades, and between a mature teratoma and a somatically transformed malignant neoplasm within the same tumor in the same patient [8]. In principle, these experiments validate a universal cancer theory based on pertinent clinical observations. In practice, they may have proven a unified theory of cancer with immense clinical implications—rendering curable TGCT even more curable, and most incurable solid tumors more treatable, if not curable.

## 2. Stem Cell Theory of Cancer

Nowadays, we attribute a stem cell theory of cancer to Rudolf Virchow (Figure 1). He was the proponent of a famous doctrine, *Omnis cellula e cellula* (i.e., every cell from a cell) [9]. Just as an animal can spring only from an animal and a plant only from a plant, a cell must arise from a previously existing cell. However, the question remains whether a cancer cell originates from a normal cell, and if so, what kind of normal cell.

In 1863, Virchow began a series of lectures about cancer’s arising from a reservoir of undifferentiated cells in the connective tissue of various organs [10]. He proposed that these embryonic cells (i.e., stem cells) were multipotent and had the capacity to generate different lineages or types of cancer.

Virchow was also the first pathologist to describe a subtype of TGCT, teratoma—from the Greek *teras*, or “monster” [10]. Within teratomas, we sometimes find terminally differentiated tissues such as a tooth, a piece of bone, brain cells, gland or gut tissue, or strands of hair. In an adult human, teratomas are usually admixed with other, less well-differentiated germ-cell tumor types, such as embryonal carcinoma (whose cells resemble early embryonic cells) and/or yolk sac tumor and choriocarcinoma (whose cell types resemble those of an embryo’s support structures, the yolk sac and placenta, respectively).

Another giant in the annals of a stem cell theory of cancer is Leroy Stevens (Figure 2). One day, he noticed a mouse with a large scrotum. It turned out that there was a tumor in its testis. About 2 percent of mouse strain 129 would spontaneously develop testicular teratomas [11]. Unbeknownst to Stevens, his experiments on mice of strain 129 would lay the groundwork for modern day stem-cell research.

In 1964, Stevens demonstrated that stem cells derived from the gonadal ridge of a developing mouse 129 embryo, when grafted onto an adult mouse testis, converted into embryonal carcinoma [11]. Subsequently, Ivan Damjanov and Davor Solter (1974) performed similar experiments and reported that normal stem cells derived from a mouse embryo of a strain not prone to forming teratomas became malignant cells, too [12]. Conversely, Karl Illmensee (1978) showed that embryonal carcinoma cells inserted into the inner cell mass of a mouse blastocyst (implantation-stage embryo) behaved like normal stem cells and became part of a normal mosaic mouse [13].

Evidently, normal stem cells and malignant stem cells are interchangeable depending on whether they reside in a stem-ness microenvironment or an onco-niche, respectively.

Stevens also discovered that different germ cells in a stem cell hierarchy have different malignant potential. He proved a stem cell rather than a genetic origin of cancers by showing that mice lacking primordial germ cells failed to develop teratoma despite carrying the putative genetic mutation, i.e., Steel (*SL*), which would have caused teratoma [14,15].

Perhaps Barry Pierce resurrected the stem cell theory of cancer in the modern era. In 1994, he and Stewart Sell published, “Maturation arrest of stem cell differentiation is a common pathway for the cellular origin of teratocarcinomas and epithelial cancers” [16]. This influential paper revived the idea that cancer has a stem cell origin and is a stem cell disease. Again, TGCT served as an invaluable tumor model to test the idea whether a poorly differentiated tumor differentiates versus whether a well-differentiated tumor dedifferentiates.

According to Sell and Pierce, when progenitor stem-like cells “fail to differentiate normally and instead accumulate (i.e., they undergo maturation arrest),” cancer growth ensues. However, we may interpret the resultant abnormal “halfway” cells as progenitor stem-like cells in suspended differentiation or progeny somatic cells in reversed differentiation (i.e., dedifferentiation or reprogramming). Inevitably, it is a challenge to prove an arrest in differentiation versus an occurrence of dedifferentiation unless we have a pertinent hypothesis (theory) in place and proper tumor model available to perform the experiments and test the hypothesis in question.

## 3. Unified Theory of Cancer

A unified theory of cancer must have universal appeal and application. Because TGCT is a relatively simple and straightforward stem cell tumor, we have thus far proved that it is a stem cell disease with relatively crude but ingenious experiments. It is a matter of time before we demonstrate the same results using modern advanced technology in a variety of other tumor types. Several such experiments come to mind.

In 2004, Timothy Wang and colleagues performed a provocative experiment to prove a stem-cell origin of cancer [17]. They demonstrated that an epithelial malignancy could have arisen from bone marrow-derived stem cells. Interestingly, cancer developed in the stomachs of a common strain of lab mice only after induction of protracted, chronic *Helicobacter* infection. It did not form after brief injury, acute inflammation, or transient cell loss. A major implication of this study is that for a cancer to form, it does not matter what kind of stem cell it is or where the stem cell originates, as long as it is a stem cell. One should never underestimate the universality and plasticity of stem cells and their capacity to form diverse cancers throughout the body. This seminal experiment provided irrefutable evidence that stem cells play a central role in carcinogenesis and that cancer is a stem-cell disease.

Nowadays, many scientists design experiments to elucidate a genetic origin of cancer by manipulating a selected set of genetic pathways in a particular subset of cells in the targeted organ. Unfortunately, it is hard to trace the effects of any single factor through a maze of redundant pathways and complex networks. It is also difficult to track the fate of specific cell types across time and space during a dynamic process such as carcinogenesis.

Philip Beachy and colleagues (2014) overcame this potential shortcoming with a model of chemical carcinogenesis in bladder cancer [18]. The team used genetic tools to mark different cell types in the bladders of live mice with distinct fluorescent colors, then exposed the mice for several months to a mutagenic component of cigarette smoke—“the most important known risk for human bladder cancer”. Tracing back the lineage of the muscle-invasive bladder carcinomas that developed, they discovered that these cancers arose exclusively from *Shh*-expressing stem cells in the basal layer of the urothelial lining. Notably, identically exposed mice that had had only these particular stem cells ablated from their bladders did not develop cancers at all.

Beachy et al. showed that *Shh*-expressing basal cells within this precursor lesion were the de facto tumor-initiating cells, yet *Shh* expression itself was lost in their progeny, the mature tumor cells of subsequent carcinomas. Thus, gene expression in the bulk of a tumor’s cells may be significantly different from that in their ‘parent’ cancer-initiating cell.

An unsolvable puzzle that has forever baffled scientists: Why do so many cells with supposed ‘cancer genes’ never become cancerous? Often enough, researchers find cancer-associated gene mutations in perfectly normal, healthy cells that never become cancer cells.

Leonard Zon and colleagues (2016) may have untied the Gordian knot of cancer genes by performing a consummate experiment in the zebrafish [19]. They traced the origin of malignancy to stem-ness and indisputably proved that cancer is a stem-cell disease.

The zebrafish *crestin* gene is expressed in embryonic neural crest progenitor cells, the stem cells that give rise to pigment-producing melanocytes, among other cell types. *Crestin* becomes undetectable by the time the zebrafish larva hatches, but is re-expressed in induced melanoma tumors of adult fish.

Zon et al. demonstrated that within a “cancer-ized field” of melanocytes, all carrying the *BRAF*^V600E^mutation and lacking p53 protection, it was a single neural crest progenitor cell expressing *crestin* that was the melanoma-initiating cell. Forcing the overexpression of *sox10*, a key transcription factor regulating *crestin* in melanocytes, accelerated melanoma formation.

Incredibly, this elegant study essentially and practically proved a stem-cell origin of cancer.

## 4. The Ultimate Experiment

There is no question or doubt that preeminent scientists like Wang, Beachy, Zon, and others belong to the pantheon of cancer research. However, without a unified cancer theory, the quintessential experiments designed to test any hypotheses related to a nondescript theory lose meaning and lack impact. In contrast, when we formulate a stem cell theory of cancer and postulate that cancer is a stem cell disease, we have an opportunity to prove that the theory has relevance in real patients with real cancers in the real world, i.e., in the clinics versus laboratories. Although people may disagree, proving the theory in human cancers and in cancer patients may be the ultimate experiment.

Returning to TGCT, Umbreit et al. made a serendipitous but astounding discovery that linked certain patients’ primary TGCT to a subsequent second (or third) malignancy [20,21,22] on an average of about 18 years later. Using a genetic marker, namely isochromosome 12p (i [12p]), Refs. [23,24] that is commonly detected in TGCT but not in any other cancers, they connected two separate disparate cancers in nature, space, and time.

Given the inherent migratory prowess, multipotent potential, and inert predisposition of normal progenitor stem-like cells, perhaps it is not surprising that progenitor stem-like cancer cells can (but progeny differentiated cancer cells may not) similarly mobilize to distant sites, change their identity (i.e., plasticity), and display late relapse (as demonstrated by SMN) over time.

Umbreit et al. verified results from prior studies showing that different tumor phenotypes (e.g., teratoma and somatic transformation) in the same tumor have an almost identical genetic signature due to their common clonal origin [25,26,27,28]. Although somatic transformation (the ability to differentiate to a different cellular lineage) could account for some SMN, it could not account for a majority of SMN derived from TGCT (i.e., seminomas) that do not normally undergo somatic transformation.

Therefore, although genetic changes may be king, cellular context is key. An aberrant progenitor stem-like cell will undergo somatic transformation and/or develop into a SMN, whereas a defective progeny differentiated cell does not. In other words, whether the same genetic changes occur in a progenitor stem-like cell or a progeny differentiated cell determines whether the resultant malignant cell engages in somatic transformation and/or emerges as a SMN.

Umbreit et al. also demonstrated for the first time that embedded within a bona fide, fully differentiated, mature teratoma were progenitor stem-like cells, thereby reaffirming a central biological tenet that multipotent progenitor cells (a.k.a., adult stem cells) differentiate into diverse cellular lineages, as opposed to progeny cells dedifferentiating into stem-like cells through genetic mutations within normal and malignant tissues. Incredibly, these experiments recapitulate Virchow’s original prescient hypothesis that cancer arises from a reservoir of undifferentiated cells in the connective tissue of various organs [10].

A quintessential experiment designed to test the hypothesis of a stem cell origin of cancer would be confirmation of a hierarchical order rather than a stochastic nature in the development of somatic transformation and SMN in TGCT. Hence, Umbreit et al. have an opportunity to demonstrate that primary TGCT with a predisposition to undergo somatic transformation or evolve into a SMN also tend to harbor progenitor stem-like cells compared with those that do not (Figure 3). If mature teratomas “are not created equal”, then we will have performed another ultimate experiment showing a hierarchical order and proving that TGCT has a stem cell origin and is a stem cell disease.

## 5. Maturation Arrest Revisited

In many respects, another fundamental biological process that makes or breaks the theory of a stem cell origin or a genetic origin of cancer besides hierarchy versus stochasticity is differentiation versus dedifferentiation, respectively.

Importantly, when it concerns maturation arrest, differentiation may be at fault and dedifferentiation is moot.

Although the idea of differentiation versus dedifferentiation in cancer may seem subtle and trivial, the distinction is actually pivotal to our thinking and understanding of cancer.

In 2006, Shinya Yamanaka and colleagues made history when they demonstrated that four stem-ness genes (namely, *Oct4*, *Sox2*, *Klf4*, and *c-myc*) were sufficient to convert certain mature cells into immature cells, thereby producing induced pluripotent stem cells (iPSC) [29].

Unbeknownst to most people, i(PSC) has a dark side. People may not realize or choose to forget that iPSC created in the laboratory have a propensity to form tumors [30,31,32,33,34]. This certainly does not bode well for the use of iPSC for regenerative medicine in the clinic. However, it is supposed to be a bonanza for those who may have wished to prove that cancer has a stem cell origin in the laboratory.

In other words, when we mess with stem cells and disrupt stem-ness by inducing improper differentiation in a progenitor stem cell during maturation arrest or inciting improper dedifferentiation in a putative differentiated cell with stem-ness genes, we initiate and promote cancer formation. A stem-cell theory of cancer gives culpable genetic defects an appropriate cellular context (if not accomplice) during carcinogenesis. It is synonymous with the concept that cancer has a stem cell origin and is a stem cell disease.

After all, iPSC is another ingenious experiment designed to test the genetic theory of cancer and the plausibility of dedifferentiation. Indeed, those same experiments would have been just as illuminating and even more enlightening had we designed them to test the stem cell theory of cancer and the inevitability of differentiation. In fact, Yamanaka had performed another supreme experiment that irrefutably proved a stem-cell theory of cancer. In which case, he would have performed the right experiment to test the wrong hypothesis.

People tend to forget that an incredible experiment is still only an experiment, and an important technical advancement is not equivalent to a conceptual breakthrough. After all, showing stem cell properties when expressing stem cell genes is what experiment is all about. It works better in certain cell types and under certain conditions. However, to think that a laboratory wonder will be a therapeutic marvel contradicts the scientific method and is contrary to our clinical practice [35].

In nature, differentiation is a well-established and accepted phenomenon. However, dedifferentiation is not a clear-cut natural observation. Certainly, most fully differentiated cells have a limited life span and tend to vanish or perish rather than regenerate or reincarnate. Perhaps someone will find a way to transmute a butterfly into a caterpillar or a frog into a tadpole in the confines of a laboratory. However, no one has yet (and probably never will) observed a butterfly transmuting into a caterpillar or a frog into a tadpole in real life.

Perhaps we should remind ourselves that we design experiments to test, not to generate, hypotheses. The observations in an experiment are the results we use to validate a specific hypothesis. However, if the hypothesis in question is erroneous to start with, because it is engendered from an improper or impertinent observation of nature and does not adhere to the scientific method, then the whole idea, hypothesis, and experiments could be misguided, and the subsequent results, misleading.

## 6. A Curable Cancer

Ultimately, the best proof of a unified theory of cancer is that it enhances the cure and improves the care of patients with a variety of cancer.

TGCT is a prototype stem cell tumor that is surprisingly curable. It is a superb tumor model for us to learn how to cure other intractable solid tumors. It is also a requisite tumor model for us to learn how to identify and treat lethal cancers in general, because of its relative simplicity (genetically), predictability (clinically), and accessibility (surgically) [36,37]. Importantly, a history of TGCT is replete with lessons about cancer cure.

Umbreit et al. put the values of clinical and basic research in their rightful places, according to the scientific method. Their clinical research using patient cases and samples investigated the value of a stem cell versus genetic theory of cancer and its respective role in personalized cancer care. Their basic research elucidated the mechanisms of action of specific treatment modalities in a curable cancer.

Their results forewarn us that although a grand cancer theory may be popular, it can be fallacious. A hypothesis may be conventional, but erroneous; the experiments flawless, but misguided; and the results convincing, but misleading. Sometimes, even a bad treatment may benefit some patients, but for the wrong reasons. Ideally, a correct cancer theory will produce effective treatments, with overarching benefits, for the right reasons.

Undoubtedly, our thinking and understanding affect the diagnosis, prognosis, and treatment of cancer. We formulate different hypotheses of cancer according to our observations and perform the necessary experiments to test the respective hypotheses. The ensuing experimental results are bound to be self-serving, if not self-fulfilling. The final judgement should be whether the treatments based on our respective hypotheses actually deliver real tangible cancer cures or mere marginal clinical improvements.

Umbtreit et al. demonstrated that although genetic makeup is important, cellular context is paramount. Their research results constitute a major paradigm shift because they overthrow traditional thinking about a genetic versus cellular origin of cancer and overturn conventional acceptance of the role of targeted therapy and precision medicine in cancer care.

Hence, the same genetic defect has different effects in a progenitor versus progeny cells. Targeted therapy (unlike multimodal therapy) provides limited and temporary clinical benefits because we aim for certain genes but miss the whole cancer. Precision medicine (in contrast to integrative medicine) is unlikely to be curative, because cancer itself can be rather imprecise—dynamic not static, complex not simple, interactive instead of isolated, and integrated rather than separated.

TGCT teaches us that to cure cancer, we need to acquire and apply proper biological insights (e.g., genetic defects versus cellular contexts) and clinical acumen (e.g., drug development versus therapy development). For example, we have learned that acquisition and accumulation of genetic mutations may not be at play when SMN and late recurrent tumors do not acquire nor accumulate more mutations over time. After all, mutation is less likely to occur in a dormant progenitor stem cell and more likely to disappear in a short-lived aberrant differentiated cell. Furthermore, we have learned how to use the same drugs in the right patients under the right situations. Hence, for patients with early seminoma, we radiate it, and for those with residual teratoma, we remove it. In fact, to improve the cure rate of patients with stage I TGCT, we paradoxically should use less chemotherapy, i.e., fewer drugs.

## 7. Conclusions

When we aspire to cure cancer, we may need to search no further than a curable cancer, such as TGCT. After all, a germ cell is a prototype stem cell. Importantly, TGCT provides a classic stem cell model of cancer that teaches us some invaluable lessons about curing an intractable cancer, in all its revelations and manifestations.

In a multicellular organism, normal tissues contain progenitor stem cells and progeny differentiated cells to assure both continuity and specialty, diversity and community, plasticity and stability. In a multicellular malignant tumor, a stem cell theory of cancer unites genetics with epigenetics; a unified theory of cancer connects all the cancer hallmarks with the various compartments, all the different components, and the ubiquitous microenvironment of cancer.

We pay tribute to the legacy of Virchow, Stevens, Pierce, and other trailblazers who elucidated and established the doctrine that TGCT has a stem cell origin. We acknowledge the seminal work of Wang, Beachy, Zon, and other pioneers who resurrected and revitalized the concept that cancer is a stem cell disease.

We demonstrated that this idea of TGCT having a stem cell origin, and of cancer being a stem cell disease, is alive in both the clinics and in the laboratory in real patients and in human tumors. Indeed, to prove a stem cell origin of cancer in patients with TGCT and their SMN, in TGCT with somatic transformation, and in a variety of disparate mature teratomas after chemotherapy is an incredible experiment of nature, if not of man.

We have learned from TGCT that to perform proper scientific research we need to adopt and adhere to the proper scientific method. Ultimately, the best scientific proof supporting a stem cell theory of cancer is that it has clinical relevance by enhancing the cure and improving the care of not just patients with TGCT but also those with other currently intractable solid tumors.

Finally, to cure and to take better care of those patients with heterogeneous rather than homogeneous cancers, mixed rather than pure tumors, complex rather than simple neoplasms, perhaps we should reconsider and revisit whether it is more beneficial to practice the often-neglected multimodal therapy than the much-touted targeted therapy and provide singular integrated medicine than popular precision medicine.

This is the essence of a stem cell origin of cancer. These are the lessons from a prototype curable cancer.

## Figures and Tables

**Figure 1 cancers-13-00660-f001:**
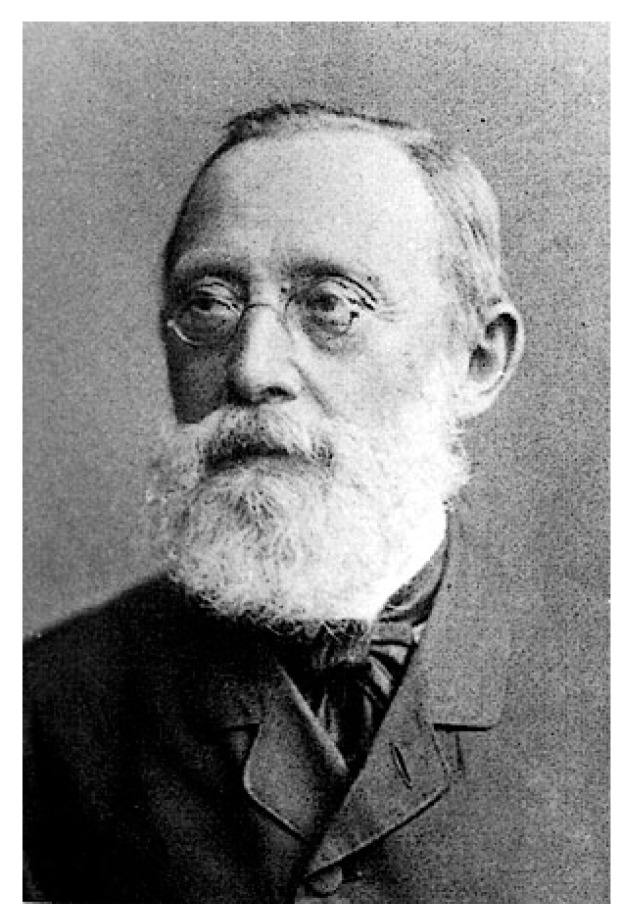
“Rudolf Ludwig Karl Virchow,” photographed by J. C. Schaarwächler in 1891, reproduced with permission from the Wellcome Medical Museum, London, UK.

**Figure 2 cancers-13-00660-f002:**
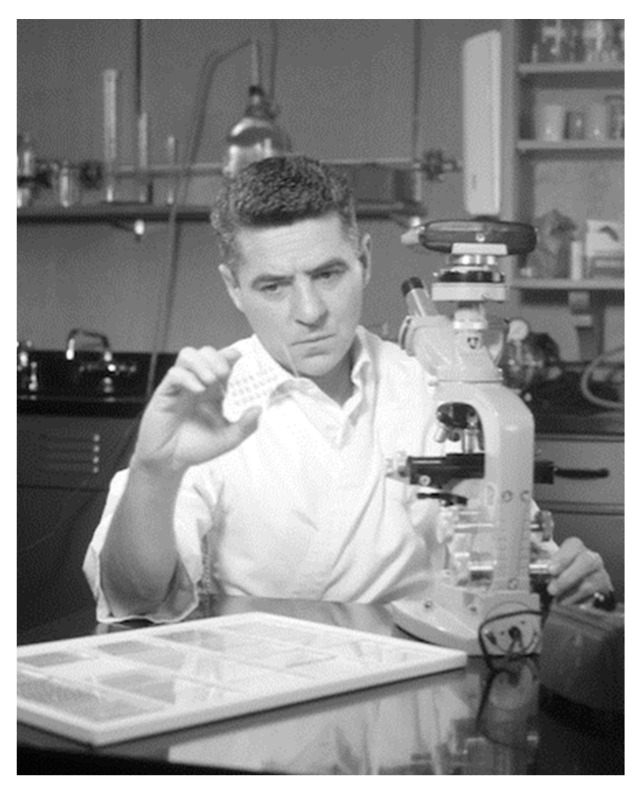
Leroy C. Stevens. Reproduced with permission from The Jackson Laboratory, Bar Harbor ME.

**Figure 3 cancers-13-00660-f003:**
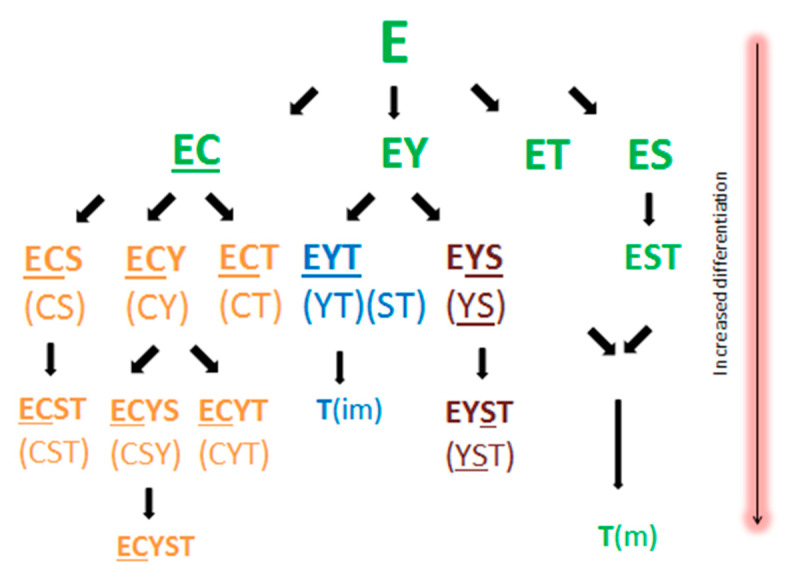
Subtypes of nonseminomatous mixed germ cell tumor of the testis. Courtesy of *Cancer* published by Wiley Periodicals, Inc. on behalf of American Cancer Society.

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
