# Peer review of "Curing Cancer: Lessons from a Prototype"

_cancers, 2021, doi:10.3390/cancers13040660_

Round 1
Reviewer 1 Report
The author can apply stem cell theory and unified theory from other cancer to TGCT. It is very impressive concept about TGCT.
Author Response
Thank you for your succinct and positive comments.
Reviewer 2 Report
In the perspective article entitled “Curing cancer: lessons from a prototype”, the authors expressed their opinions about cancer origins. Starting with germ cell tumor of the testis (TGCT), the authors presented multiple examples to explain the importance of stem cells in carcinogenesis. Overall, the manuscript is well written.
Some minor concerns: It was mentioned in the "simple summary" on page 1 “This complete reversal in the cure rate of TGCT is not because we have designed better drugs (we have not), but because we have learned how to use the same drugs in the right patients under the right settings.” It is not clear how the same drugs, when used “in the right patients under the right settings” have improved the patients’ outcomes. It will be helpful if the authors elaborate a little more on this aspect. Also, the authors cited some seminal work from previous studies to support that cancer is a stem cell disease rather than a genetic disease. It would be helpful if the authors summarize the studies or show how the conclusions were drawn. For example, on page 3, Barry Pierce’s work was cited without a description of how the conclusion “cancer has a stem cell origin and is a stem cell disease” was reached.
Reviewer 3 Report
The authors showed a review and perspective article regarding stem cell theory of cancer using germ cell tumor of the testis. The article is a very interesting well-designed and well-written article. No further comments from my side.
Author Response

(The authors gave the same response as above.)
